# Learning to predict RNA sequence expressions from whole slide images with applications for search and classification

Areej Alsaafin[1,2], Amir Safarpoor [2], Milad Sikaroudi[2], Jason D. Hipp[3] & H. R. Tizhoosh [1,2✉]

Deep learning methods are widely applied in digital pathology to address clinical challenges such as prognosis and diagnosis. As one of the most recent applications, deep models have also been used to extract molecular features from whole slide images. Although molecular tests carry rich information, they are often expensive, time-consuming, and require additional tissue to sample. In this paper, we propose tRNAsformer, an attention-based topology that can learn both to predict the bulk RNA-seq from an image and represent the whole slide image of a glass slide simultaneously. The tRNAsformer uses multiple instance learning to solve a weakly supervised problem while the pixel-level annotation is not available for an image. We conducted several experiments and achieved better performance and faster convergence in comparison to the state-of-the-art algorithms. The proposed tRNAsformer can assist as a computational pathology tool to facilitate a new generation of search and classification methods by combining the tissue morphology and the molecular fingerprint of the biopsy samples.

[1] Rhazes Lab, Artificial Intelligence and Informatics, Mayo Clinic, Rochester, MN, USA. [2] Kimia Lab, University of Waterloo, Waterloo, ON, Canada. [3] Division of Computational Pathology and AI, Mayo Clinic, Rochester, MN, USA. ✉email: tizhoosh.hamid@mayo.edu

Pathologists use histopathology to diagnose and grade cancer after examining a biopsy specimen. The introduction of digital pathology, advances in computing technology, and the expanding availability of massive datasets made it possible to train increasingly complex deep learning models for various clinical tasks. Convolutional neural networks (CNNs) surpassed all other traditional computer vision algorithms in a wide range of clinical applications, including cancer subtyping[1], whole-slide image (WSI) search and categorization[2], mitosis detection[3], and grading[4], among deep learning architectures.

However, there have been a few attempts to connect the morphological characteristics embedded in the images to molecular signatures, recently[5–8]. For instance, recent research has revealed that statistical models can link histomorphological traits to mutations in organs, including the lung and prostate[9,10]. Mutations and epigenomic modifications are known to cause large variations in gene expression. Therefore, characterization of the gene expression can be vital for diagnosis and treatment[11]. Even though more affordable whole transcriptome sequencing tools for studying gene information have been established, they are still a long way from being widely used in medical centers[12]. On the other hand, the recovery of molecular features from hematoxylin & eosin (H&E) stained WSIs is one of the faster and less expensive options. The capability to predict gene expression using WSIs, either as an intermediate modality or as an outcome, has been demonstrated to aid diagnosis and prognosis[5,8]. Previous studies have drawn attention to gene expression prediction using WSI; however, the size of WSIs and the amount of well-annotated data still impose serious challenges. In particular, sample selection and WSI representation is an open topic that is often handled arbitrarily.

According to the most recent global cancer statistics report, in 2020, there were an estimated 431,288 new cases of kidney cancer and 179,368 deaths globally[13]. The renal cell carcinoma (RCC) is the most common kidney cancer that is responsible for 85% of malignant cases[14]. From a single malignant phenotype to a heterogeneous group of tumors, our knowledge about RCC has evolved over time[14]. Among all RCC histologic subtypes, ccRCC, pRCC, and crRCC make almost 75%,16%, and 7% of the whole RCC cases, respectively[14]. RCC subtypes differ in their histology, molecular characteristics, clinical outcomes, and therapeutic responsiveness as a result of this heterogeneity. For instance, because the 5-year survival rate differs across different subtypes, proper subtype diagnosis is critical[15]. All methods in this work are applied on RCC slides to identify the subtypes using search and classification.

Here, we introduce tRNAsformer (pronounced *t*-RNAs-former), a deep learning model for end-toend gene prediction and learning WSI representation at the same time (Fig. 1 and Supplementary Fig. 1). Our model employs transformer modules built on the attention mechanism to gather information required for learning WSI representations. The attention-based mechanism allows learning information that attributes to some specific features in the image and scores them against other features. In doing so, the model would capture how a feature relates to the others in the image so that it focuses on the relevant part of the image. Moreover, tRNAsformer employs the concept of multiple instance learning (MIL)[16] to handle the problem of having the real gene expression values per WSI instead of per tile. MIL is a form of weakly supervised learning where training instances are arranged in bags (sets), and a label is provided for the entire bag. To train our model, we used data from The Cancer Genome Atlas (TCGA) public dataset to gather kidney WSIs and their related RNA-seq data. For WSIs, we presented our findings related to gene prediction and internal representation. Finally, we tested the generalization of our model in terms of learned WSI internal representation against state-of-the-art benchmarks using an external kidney cancer dataset from the Ohio State University.

## Results

In this section, we assess the performance of tRNAsformer in terms of the two main tasks that it has been trained for: gene expression prediction from WSI and WSI representation for image search and classification. The performance of tRNAsformer in predicting gene expressions has been compared with the performance of one of a state-of-the-art model, called HE2RNA. The performance of tRNAsformer in terms of learning rich information to represent WSIs has been compared to two other methods, namely Yottixel and Low Power.

**A model for predicting gene expression from WSIs**. The FPKM-UQ files containing 60,483 Ensembl gene IDs were utilized in this study[17]. During the preprocessing step (described in section "Gene expression preprocessing"), some of the gene expression values were selected and then transformed first.

Both models, tRNAsformer and HE2RNA, were compared for three different criteria, namely mean correlation coefficient of predictions, the number of genes predicted significantly better than a random baseline, and the prediction error. In the first experiment, the correlation is assessed for each gene separately using Pearson and Spearman's correlation coefficient. If the datasets are normally distributed, the Pearson correlation coefficient measures the linear connection between them. The Pearson correlation coefficient varies between $-1$ and $+1$. A correlation of $-1$ or $+1$ denotes a perfect linear negative or positive relationship, respectively, whereas a correlation of 0 denotes no correlation. The *p*-value roughly represents the probability that an uncorrelated system can produce datasets with a Pearson correlation at least as high as the one calculated from these datasets. The Spearman correlation, unlike the Pearson correlation, does not require that both datasets be normally distributed. Figure 2 displays the distribution of the correlation coefficient for 31,793 genes predicted by different models.

Figure 1 illustrates the distribution of the correlation coefficients for 31,793 genes predicted by different models along with their true values in the test set of TCGA. As seen in Fig. 2, the mean correlation coefficient $R$ grew with depth from $L = 1$ to $L = 8$. The mean $R$ value declines after eight blocks of Transformer encoders, suggesting that increasing the number of layers does not enhance gene expression predictions. In terms of the correlation of the predicted gene expressions with real values, tRNAsformer models from $L = 2$ to $L = 8$ achieved comparable results with a slight improvement as compared to HE2RNA. Beyond correlation values, the literature uses violin plots[18–21] because the large number of data points per patient drastically reduces the visibility of any interpretable clues if other methods such as scatter plots[22] are used.

Pearson and Spearman's correlation coefficients and *p*-values were computed between the predicted and the true value of the gene expression for each gene. Two multiple-hypothesis testing methods, namely Holm–Šidák (HS) and Benjamini–Hochberg (BH), were utilized to adjust the *p*-values. If the *p*-value of the $R$ coefficient was less than 0.01 after correction for multiple-hypothesis testing, the prediction was significantly different from the random baseline[23,24]. Similar to ref. 5, multiplehypothesis testing was done using both HS and BH correction. The results are shown in Table 1 for all architectures.

As it is demonstrated in Table 1, increasing the depth of the tRNAsformer from one to eight increases the number of genes that are significantly different from a random baseline. Similar to the results in Fig. 2, there is a decrease in the number of genes when the

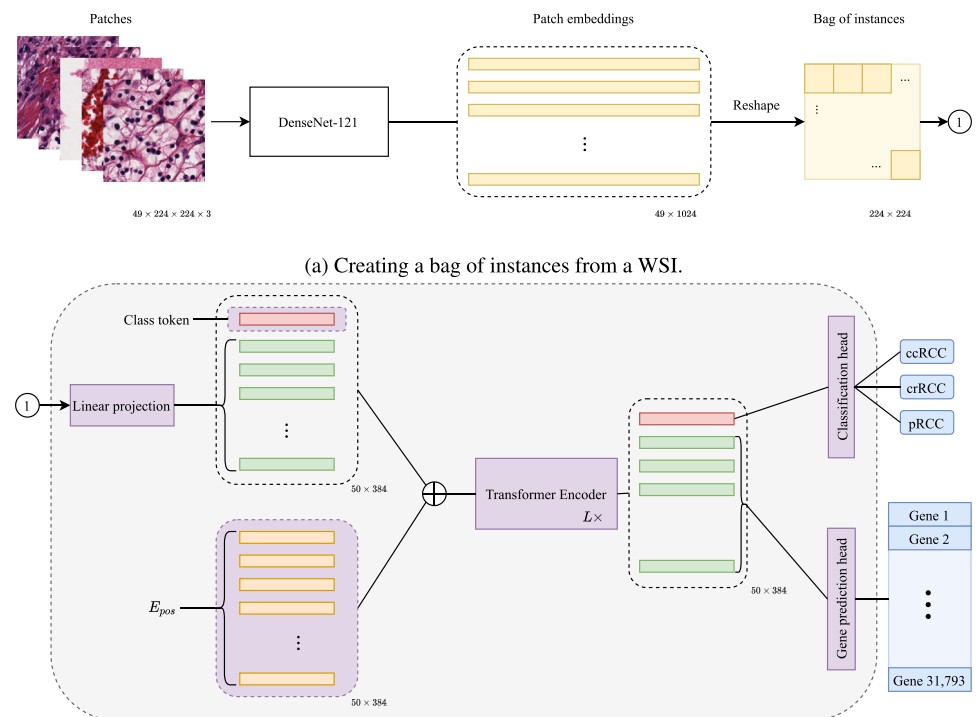

(a) Creating a bag of instances from a WSI.

(b) Internal schematic of how data flows in tRNAsformer.

**Fig. 1 A diagram showing how tRNAsformer works. a** 49 tiles of size 224 × 224 × 3 selected from 49 spatial clusters in a WSI are embedded with a DenseNet-121. The outcome is a matrix of size 49 × 1024 as DenseNet-121 has 1024 deep features after the last pooling. Then the matrix is reshaped and rearranged to 224 × 224 matrix in which each 32 × 32 block corresponds to a tile embedding 1 × 1024. **b** Applying a 2D convolution with kernel 32, stride 32, and 384 kernels, each 32 × 32 block has linearly mapped a vector of 384 dimensional. Next, a class token is concatenated with the rest of the tile embeddings, and Epos is added to the matrix before entering L Encoder layers. The first row of the outcome, which is associated with the class token, is fed to the classification head. The rest of the internal embeddings that are associated with all tile embeddings are passed to the gene prediction head. All parts with learnable variables are shown in purple.

depth reaches 12 blocks of the Transformer Encoder. On the other hand, the model based on the design of HE2RNA scored inferior to nearly all other tRNAsformer models (except for $L = 1$).

We selected MAE, RMSE, and RRMSE[25] to calculate the error between the prediction and real gene expression values. MAE, RMSE, and RRMSE are defined as

$$
\mathrm{MAE} = \frac{\sum_{(x_i, y_i) \in D_{\text{test}}} |\hat{y}_i - y_i|}{|D_{\text{test}}|}, \tag{1}
$$

$$
\mathrm{RMSE} = \sqrt{\frac{\sum_{(x_i, y_i) \in D_{\text{test}}} (\hat{y}_i - y_i)^2}{|D_{\text{test}}|}}, \tag{2}
$$

$$
\mathrm{RRMSE} = \sqrt{\frac{\sum_{(x_i, y_i) \in D_{\text{test}}} (\hat{y}_i - y_i)^2}{\sum_{(x_i, y_i) \in D_{\text{test}}} (\bar{y} - y_i)^2}}, \tag{3}
$$

where $D_{\text{test}}$ denotes the test set, $(x_i, y_i)$ is the $i$-th sample $x_i$ with ground truth $y_i$, $\hat{y}_i$ is the predicted value of $y_i$, $\bar{y}$ is the mean value over the targets in the test set, and $|D_{\text{test}}|$ is the number of samples in the test set. The results are given in Table 2.

Similar to the results in Fig. 2 and Table 1, increasing the number of Transformer Encoder blocks from eight to 12 significantly degrades the performance of the model. The correlation values achieved by tRNAsformer are comparable to the values of the HE2RNA model.

The hyperparameters of both tRNAsformer and HE2RNA models were optimized before conducting the experiments. HE2RNA uses all the tiles of a WSI to train the model and to produce a prediction for every tile. This helps in improving the

error rate when averaging a large number of tile predictions to get one prediction per slide. Averaging multiple predicted values (tile predictions) would increase the chance of having a more similar value to the real value as the effect of applying this method is like averaging the error rate of all the predictions to get a single representative value of all the tiles. However, producing a gene expression score per tile, like HE2RNA, results in ignoring the dependencies between the tiles of a WSI as the real values are per WSI not per tile. tRNAsformer solves this issue by treating a WSI in its entirety and therefore producing one prediction per WSI. The model employs the concept of multi-instance learning to handle the problem of having the real gene expression values per WSI instead of per tile. Furthermore, from a computational point of view, considering all the tiles to train the network is prohibitively time and resource-consuming as a single WSI can easily have several thousands of tiles. Therefore, in tRNAsformer we addressed this issue by incorporating the attention mechanism and multiple instance learning concept in the training process.

Overall, as it can be noted from the above results, the performance of tRNAsformer models with L = 2 to 8 are comparable. However, by considering all the metrics used to evaluate the models, tRNAsformer with $L = 4$ performs the best. In this paper, we are presenting the performance of tRNAsformer with different depths as the model depth can be selected based on the available resources. For example, in the case of limited resources, $L = 2$ can be used as it can achieve comparable performance as the deeper models but with less resource requirement.

**Transcriptomic learning for WSI representation - WSI classification**. The classification experiments were conducted to assess

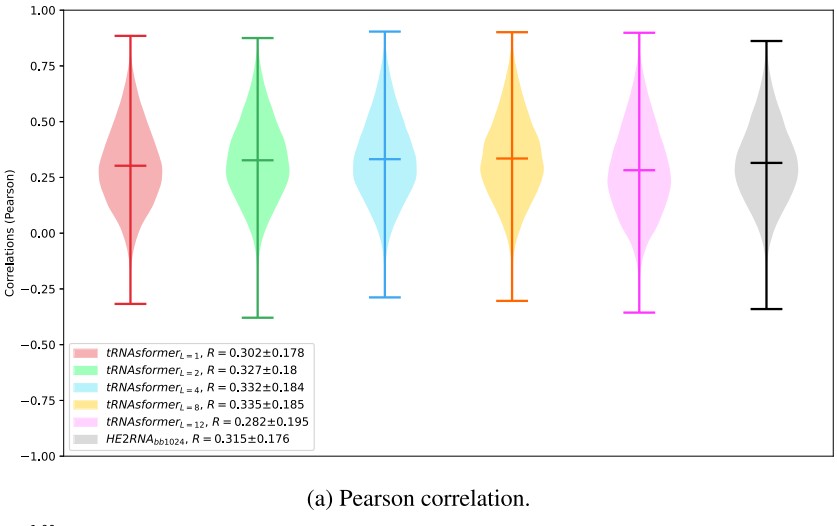

(a) Pearson correlation.

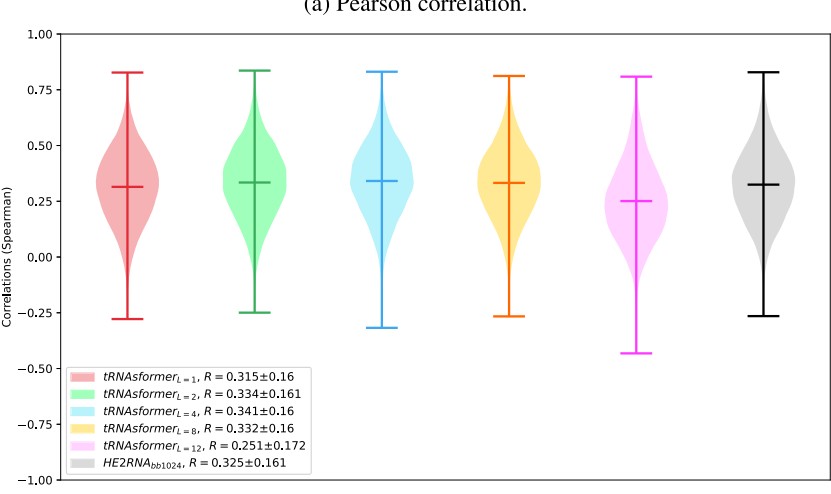

(b) Spearman correlation.

**Fig. 2 The distribution of the correlation coefficients between 31,793 genes predicted and their true value for the TCGA test set.** The violin diagrams depict the distribution, min, max, and mean values of the correlation coefficients. **a** Violin diagrams for Pearson correlation coefficients and **b** violin diagrams for Spearman's correlation coefficients. The violin diagrams are plotted for tRNAsformer$_L$ for $L = (1, 2, 4, 8, 12)$ and HE2RNA$_{bb1024}$. The mean and standard deviation of the correlation coefficients are included in the legend for violins from left to right.

**Table 1 The number of genes was predicted with a statistically significant correlation ($p$-value < 0.01) under HS and BH correction.**

| Model | Pearson | | Spearman | |
|---|---|---|---|---|
| | HS | BH | HS | BH |
| tRNAsformer$_{L=1}$ | 29,990 | 30,797 | 30,427 | 31,042 |
| tRNAsformer$_{L=2}$ | 30,338 | 31,014 | 30,695 | 31,141 |
| tRNAsformer$_{L=4}$ | 30,433 | 30,996 | 30,858 | 31,266 |
| tRNAsformer$_{L=8}$ | 30,344 | 31,002 | 30,741 | 31,181 |
| tRNAsformer$_{L=12}$ | 28,933 | 30,187 | 28,938 | 30,210 |
| HE2RNA$_{bb1024}$ | 30,249 | 30,937 | 30,663 | 31,163 |

The total number of predicted genes is 31,793. These values are computed using the TCGA test dataset.

**Table 2 Prediction error for tRNAsformer and HE2RNA$_{bb1024}$ models quantified by MAE, RMSE, and RRMSE.**

| Model | MAE | RMSE | RRMSE |
|---|---|---|---|
| tRNAsformer$_{L=1}$ | 1.31 ± 1.04 | 1.67 ± 1.20 | 1.02 ± 0.16 |
| tRNAsformer$_{L=2}$ | 1.30 ± 1.03 | 1.65 ± 1.17 | 1.02 ± 0.16 |
| tRNAsformer$_{L=4}$ | 1.30 ± 1.08 | 1.63 ± 1.19 | 0.98 ± 0.11 |
| tRNAsformer$_{L=8}$ | 1.37 ± 1.02 | 1.69 ± 1.13 | 1.11 ± 0.27 |
| tRNAsformer$_{L=12}$ | 1.50 ± 1.10 | 1.79 ± 1.26 | 1.10 ± 0.43 |
| HE2RNA$_{bb1024}$ | 1.29 ± 1.08 | 1.63 ± 1.20 | 0.96 ± 0.08 |

All errors are calculated using the TCGA test set.

the quality of internal representation learned by the proposed model. To begin, 100 bags have been created from each TCGA test WSIs. According to Supplementary Table 1, a total of 8000 bags were created from TCGA test set, as there were 80 WSIs. The same models that were trained in the previous section to predict RCC subtypes were assessed for the classification task as well. The

accuracy, macro, and weighted F1 scores are presented for all models in Table 3. The confusion matrices of different models are displayed in Supplementary Fig. 2. All values reported here are based on slide-level classification results. The prediction is made for all bags in order to calculate slide-level values. Each test slide's label prediction is chosen as the most common prediction among all bags created from that slide. The WSI representations learned by the models are projected onto a plane created by the first two

**Table 3 The accuracy, macro, and weighted F1 scoes for classification on the TCGA test set and the external dataset for all classification models.**

| Model | TCGA | | | External dataset | | |
|---|---|---|---|---|---|---|
| | | F1 score | | | F1 score | |
| | Accuracy | Macro | Weighted | Accuracy | Macro | Weighted |
| tRNAsformer$_{L=1}$ | 93.75% | 0.9488 | 0.9366 | 82.39% | 0.8241 | 0.8223 |
| tRNAsformer$_{L=2}$ | 95.00% | 0.9406 | 0.9496 | 81.69% | 0.8161 | 0.8145 |
| tRNAsformer$_{L=4}$ | 96.25% | 0.9511 | 0.9625 | 78.87% | 0.7899 | 0.7871 |
| tRNAsformer$_{L=8}$ | 95.00% | 0.9414 | 0.9502 | 82.39% | 0.8251 | 0.8227 |
| tRNAsformer$_{L=12}$ | 92.50% | 0.9392 | 0.9243 | 80.28% | 0.8072 | 0.8034 |
| Low power method[29] | 93.75% | 0.9488 | 0.9366 | 73.76% | 0.7388 | 0.7385 |

principal components found using PCA to depict the internal representation of our models in two-dimensional space. The two-dimensional PCA projections are shown in Supplementary Fig. 3.

Because of variations in hospital standards and methods for tissue processing, slide preparation, and digitization protocols, the appearance of WSIs might vary significantly. As a result, it is important to ensure that models built using data sources are resistant to data-source-specific biases and generalize to real-world clinical data from sources not used during training[26]. For testing the generalization of our trained models, 142 RCC WSIs are used from the Ohio State University as an independent test cohort (see Section "The Ohio State University kidney dataset").

First, 100 bags were created from each external test WSIs. According to Supplementary Table 1, a total of 14,200 bags were created from the TCGA test set, as there were 142 WSIs. The same models that were trained in the previous section to predict RCC subtypes are used to report classification results for the external dataset. The accuracy, macro, and weighted F1 scores are reported for all models in Table 3. As shown in Table 3, the accuracy of tRNAsformer decreased by about 13% for the external validation. These results still show a reasonable performance, especially when considering the performance of its counterpart, which showed about a 20% decrease in accuracy. Lack of generalization due to overfitting, bias, and shortcuts is a general problem in deep learning[27,28]. However, applying more sophisticated preprocessing may improve the model performance and lead to better sensitivity when using an external dataset. The model performance can also be improved by training it on a larger dataset. However, for sake of reproducibility, we are limited to the number of WSIs available on TCGA. Furthermore, we can only consider WSIs where RNA-seq profiles were available in TCGA. The confusion matrices of different models are displayed in Supplementary Fig. 4. The WSI representations learned by the models are projected onto a plane created by the first two principal components found using PCA to depict the internal representation of the models in two-dimensional space. The two-dimensional PCA projections are shown in Supplementary Fig. 5. Supplementary Figs. 3, 5 show how well the WSI representations extracted from the tRNAsformer model can be distinguished across different classes. In other words, the figures illustrate the discriminative power of the features learned by each tRNAsformer model.

The suggested model in ref. [29], also known as the "Low Power" technique, outperformed all tilebased and state-of-the-art WSI-level approaches. The "Low Power" method's accuracy, F1 score (macro and weighted), and AUC were 73.76%, 0.7388, 0.7385, and 0.893, respectively. As it is demonstrated in Table 3 and Fig. 3, all tRNAsformer models surpass the method described in ref. [29] in all measures, namely accuracy, F1 score (macro and weighted), and AUC. In addition, as it is depicted in Supplementary Fig. 4, the tRNAsformer models tend to have more balanced correct predictions for all classes as there is a crisp diagonal line highlighted in confusion matrices. To put it another way, tRNAsformer models are good at distinguishing between all classes.

**Transcriptomic learning for WSI representation - WSI search.** WSI search experiments were conducted to assess the quality of the internal representation of the tRNAsformer. The model is tested on both TCGA and an external dataset. As it was mentioned earlier 100 instances were created from each WSI in the TCGA dataset; the TCGA test set contained 8000 instances associated with 80 slides. After training tRNAsformer, it was utilized to extract features (embeddings). To quantify the performance of tRNAsformer in WSI search, first, 100 subsets of instances were created from 8000 TCGA test instances. Next, a pairwise distance matrix is computed using the WSI embeddings (feature vectors) for each subset. The Pearson correlation is employed as the distance metric. Following the leave-one-patient-out procedure, the top-k samples were determined for each instance (WSI). Later, P@K (Precision@K) and AP@K (Average Precision@K) were computed for each subset. P@K reflects how many relevant images are present in the top-k recommendations that the model suggests, while AP@K is the mean of P@i for i = 1,…,K. Finally, the MAP@K (Mean Average Precision@K) value was computed by taking the average of 100 queries associated with 100 search subsets.

Similarly, 100 instances were created for each WSI in the external dataset. Overall, 100 subsets of 142 WSIs generated for the WSI search in the external dataset. As a result, MAP@K values were evaluated by taking an average from 100 different search experiments. The summary of MAP@K values for both the TCGA test and the external dataset are shown in Table 4.

The performance of tRNAsformer has been compared with the performance of Yottixel[30], the state-of-the-art in WSI search, in terms of the mean average precision at different k, MAP@5 and MAP@10. The MAP@5 and MAP@10 for 10 independent Yottixel runs were 0.7416 and 0.7092, respectively. tRNAsformer outperforms Yottixel in both MAP@5 and MAP@10 measures. Furthermore, tRNAsformer models provide more stability because the MAP@ K value does not drop as steeply as other search algorithms while the k increases.

## Discussion

In this paper, a multitask MIL framework based on tRNAsformer model is proposed for learning WSI representation by learning to predict gene expression from H&E slides. By incorporating the attention mechanism and the Transformer design, tRNAsformer can provide more precise predictions for gene expressions from a WSI. Meanwhile, tRNAsformer surpassed benchmarks for bulk RNA-seq prediction while having fewer hyperparameters. In

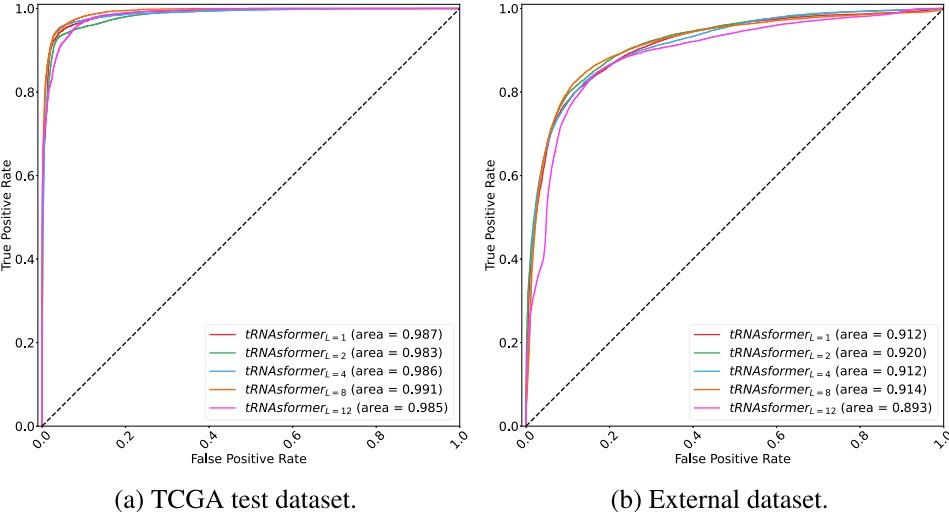

(a) TCGA test dataset.

(b) External dataset.

**Fig. 3 ROC Curves for TCGA and External Dataset.** The micro ROC curve of different models applied on **a** the TCGA test set and **b** the external dataset. The AUC is reported in the legend for all models.

**Table 4 The MAP@5 and MAP@10 values for all WSI search models applied on the TCGA test and the external dataset.**

| Model | TCGA | | External dataset | |
|---|---|---|---|---|
| | MAP@ 5 | MAP@ 10 | MAP@ 5 | MAP@ 10 |
| tRNAsformer$_{L=1}$ | 0.8966 | 0.8985 | 0.8026 | 0.8035 |
| tRNAsformer$_{L=2}$ | 0.8831 | 0.8800 | 0.7988 | 0.7976 |
| tRNAsformer$_{L=4}$ | 0.9150 | 0.9124 | 0.7819 | 0.7781 |
| tRNAsformer$_{L=8}$ | 0.9031 | 0.8996 | 0.7674 | 0.7628 |
| tRNAsformer$_{L=12}$ | 0.8762 | 0.8751 | 0.7262 | 0.7257 |
| Yottixel[2] | 0.764 | 0.717 | 0.7416 | 0.7092 |

addition, tRNAsformer learns exclusive and compact representation for a WSI using the molecular signature of the tissue sample. As a result, the proposed technique learns a diagnostically relevant representation from an image by integrating gene information in a multimodal approach.

In fact, whole slide images (WSIs) are usually labeled by treating the image in its entirety (the label is assigned to the whole image). For example, a whole slide image can be labeled as a tumor slide although it might include some normal tissue as well. Processing an entire WSI at once is not possible with present hardware technology. These images are commonly divided into smaller, more manageable pieces known as patches or tiles. However, large WSI datasets are generally softly labeled since pixel-level expert annotation is costly and labor-intensive. As a result, some of the tiles may not carry information that is relevant to the diagnostic label associated with the WSI. tRNAsformer design allowed for more efficient and precise processing of a collection of samples. It employs weekly supervised learning along with multi-instance learning (MIL) concept[16]. Weakly supervised learning is an approach to train a deep network by using the combination of the given labeled data and the weak supervision for obtaining new labeled data[31]. This approach makes training a deep network possible when the available labeled data is insufficient. Furthermore, tRNAsformer employs the concept of MIL to handle the problem of having the real gene expression values per WSI instead of per tile. MIL is a form of weakly supervised learning where training instances are arranged in bags (sets), and a label is provided for the entire bag.

A pre-trained CNN model was used for sampling and embedding image tiles before training tRNAsformer. This approach allows us to create rich intermediate embeddings from image samples as the pre-trained CNN model was trained on large image datasets. Furthermore, working with embedded sampled instances is computationally less expensive in comparison with treating each WSI as an instance. According to Supplementary Table 2, the smallest tRNAsformer model can have about 60% less hyperparameter in comparison with MLP-based model. In addition, they can be about 72% and 15% faster than MLP-based model during training and validation, respectively.

Our main aim from the comparison between tRNAsformer and HE2RNA is to demonstrate that tRNAsformer can predict gene expressions from a WSI as accurately as the state-of-the-art gene expression algorithms with simultaneously learning rich WSI representation from both morphology features as well as molecular fingerprint, which can be used for applications such as image search. tRNAsformer was able to predict gene expression scores with a slightly improved correlation compared to that achieved by HE2RNA. However, one has to bear in mind that tRNAsformer is a multi-task computational pathology tool that can be used not only for gene expression prediction but also for learning WSI representation based on the tissue morphology and the molecular fingerprint of a biopsy sample, which can be integrated into image search and classification. The correlation metric was used to evaluate only one task, which is gene expression prediction. The other task (i.e., transcriptomic learning for WSI representation for image search and classification) has been evaluated by considering an external dataset along with two other methods for comparison, namely "Yottixel" and "Low power" methods.

In contrast to ref. [7], where the spatial transcriptomics dataset was available, the proposed approach in this work uses bulk RNA-seq data. As a result, the model described in this study employs a weaker type of supervision, as it learns internal representation using a combination of a primary diagnosis and a bulk RNA-seq associated with a WSI. This is more in line with current clinical practice, which generally collects bulk RNA sequences rather than spatial transcriptomic data. Furthermore, tRNAsformer handles the problem by treating a WSI in its entirety, whereas the method explained in ref. [7] separates each tile and estimates the gene expression value for it. Therefore, the method described in ref. [7] ignores the dependencies between tiles.

Comparing to ref. [8], the proposed technique in this manuscript processes a considerably smaller set of samples with a larger field of view. In particular, the proposed technique samples bags of 49 instances of $224 \times 224 \times 3$ while the other technique[8] deployed several sampling options with at least 2500 tiles of size $32 \times 32 \times 3$ per bag. In addition, tRNAsformer learns exclusive WSI representation by learning the pixel-to-gene translation. On the other hand, none of the methodologies have an independent representation learning paradigm[5,7,8].

In conclusion, the results showed that tRNAsformer can learn reliable internal representations for massive archives of pathology slides that match or outperform the performance of cutting-edge classification and search algorithms developed[29,30]. In addition, tRNAsformer can predict gene expressions from H&E slides with comparable performance with some improvement as compared to other state-of-the-art methods[5]. We have shown that even with RNA-Seq profiles obtained from bulk cells, mostly, isolated from a different tissue section, tRNAsformer performed well in terms of predicting gene expression scores correlated to the true scores in the bulk RNA-seq profiles, which may indicate that most of the expressed genes in the tissue section used for *H&E* staining are also expressed in the tissue section used for RNA-seq quantification. However, in future research, tRNAsformer can be investigated rigorously by verifying its performance using spatial transcriptomic data in which both the RNA-seq profiling and *H&E* staining are performed on the same slice of the specimen.

## Methods

**TCGA kidney dataset**. The data used in this study was obtained from TCGA (https://portal.gdc.cancer.gov/). Only cases that have WSI as well as RNAseq profile were considered. We selected H&E-stained formalin-fixed, paraffin-embedded (FFPE) diagnostic slides. The retrieved cases included three subtypes, clear cell carcinoma, ICD-O 8310/3, (ccRCC), chromophobe type - renal cell carcinoma, ICD-O 8317/3, (crRCC), and papillary carcinoma, ICD-O 8260/3, (pRCC). For transcriptomic data, we utilized Fragments Per Kilobase of transcript per Million mapped reads upper quartile (FPKM-UQ) files. The detailed information regarding the cases is included in Supplementary Table 1. As the mean value of the FPKM-UQ data for each gene may vary significantly between different projects, both tRNAsformer and HE2RNA models have been evaluated to predict gene expression scores of FPKM-UQ data from only one project, which is TCGA. Three kidney datasets have been considered from TCGA, which are TCGA-KIRC, TCGA-KIRP, and TCGA-KICH. Furthermore, we have excluded genes with a median expression of zero to improve the interpretability of the results. The data was split case-wise into train (%80), validation (%10), test (%10) sets, respectively. In other words, each patient only belonged to one of the sets.

**Gene expression preprocessing**. The FPKM-UQ files contained 60, 483 Ensembl gene IDs. We excluded genes with a median of zero across all kidney cases and we left with 31,793 genes. Other studies have adopted the same strategy to improve the interpretability of the results[5]. We used $a \rightarrow \log_{10}(1 + a)$ transform to convert the gene expressions since the order of gene expression values changes a lot and can impact mean squared error only in the case of highly expressed genes[5].

**WSI preprocessing**. The size of the digitized glass slides may be $100,000 \times 100,000$ in pixels or even larger. As a result, processing an entire slide at once is not possible with the present technology. These images are commonly divided into smaller,

more manageable pieces known as tiles. Furthermore, large WSI datasets are generally weakly labeled since pixel-level expert annotation is costly and labor-intensive. As a result, some of the tiles may not carry information that is relevant to the diagnostic label associated with the WSI. Consequently, MIL may be suitable for this scenario. Instead of receiving a collection of individually labeled examples, the learner receives a set of labeled bags, each comprising several instances in MIL. For making bags of instances, the first step is to figure out where the tissue boundaries are. Using the algorithm described in ref. [29], the tissue region was located at the thumbnail (1.25× magnification) while the background and the marker pixels were removed. Tiles of size 14 by 14 pixels were processed using the 1.25 × tissue mask to discard those with less than 50% tissue. Note that $14 \times 14$ pixels tiles at 1.25 × is equivalent to the area of $224 \times 224$ pixels at 20× magnification.

The k-means algorithm is deployed on the location of the tiles selected previously to sample a fixed number of tiles from each WSI. The value of k was set to 49 for all experiments in this study. After that, the clusters are spatially sorted based on the magnitude of the cluster centers. The benefit of spatially clustered tiles is twofold; (1) the concept of similarity is more likely to be true within a narrow radius[32,33], and (2) clustering coordinates with two variables is computationally less expensive than high-dimensional feature vectors. The steps of the clustering algorithm are shown in Fig. 4.

**The tRNAsformer architecture**. The tRNAsformer is made of $L$ standard transformer encoder layers[34] followed by two heads, namely the classification, and the gene prediction head. Supplementary Fig. 1 depicts the architecture of the proposed method. The Transformer Encoder learns an embedding (also known as the class token) for the input by treating it as a sequence of feature instances associated with each WSI. It learns internal embeddings for each instance while learning the class token that represents the bag or WSI.

The classification head, which is a linear layer, receives the WSI representation **c**. Next, the WSI representation is projected using a linear layer to the WSI's score $\hat{y}$. tRNAsformer then uses crossentropy loss between the predicted score $\hat{y}$ and the WSI's true label **y** to learn the primary diagnosis. The use of the Transformer Encoder and the classification head enables the learning of the WSI's representation while training the model.

Considering a bag X = $[\mathbf{x}_1, \mathbf{x}_2, \ldots, \mathbf{x}_k]$, where $\mathbf{x}_i \in \mathbb{R}^d$, $i = 1, \ldots, k$ are the embedded tiles by DenseNet-121, an $L$-layer standard Transformer can be defined as

$$\mathbf{z}_0 = [(\mathbf{x}_{class}; \mathbf{x}_1\mathbf{E}; \mathbf{x}_2\mathbf{E}; \ldots; \mathbf{x}_k\mathbf{E})] + \mathbf{E}_{pos}, \qquad \mathbf{E} \in \mathbb{R}^{d \times D}, \mathbf{E}_{pos} \in \mathbb{R}^{(k+1) \times D} \quad (4)$$

$$\mathbf{z}'_\ell = \text{MSA}(\text{LN}(\mathbf{z}_{\ell-1})) + \mathbf{z}_{\ell-1}, \qquad \ell = 1, \ldots, L \quad (5)$$

$$\mathbf{z}_\ell = \text{MLP}(\text{LN}(\mathbf{z}'_\ell)) + \mathbf{z}'_\ell, \qquad \ell = 1, \ldots, L \quad (6)$$

$$\mathbf{c} = \text{LN}(\mathbf{z}_L^0), \quad (7)$$

$$\hat{y} = \text{L}(\mathbf{c}) \quad (8)$$

where MSA, LN, MLP, L, **E**, and $\mathbf{E}_{pos}$ are multi-head self-attention, layernorm, multi-layer perceptron block (MLP), linear layer, tile embedding projection, and position embedding (for more information see ref. [34]). The variables **E** and $\mathbf{E}_{pos}$ are learnable. The layernorm applies normalization over a minibatch of inputs. In layernorm, the statistics are calculated independently across feature dimensions for each instance (i.e., tile) in a sequence (i.e., a bag of tiles). The multi-layer perceptron block is made of two linear layers followed by a dropout layer. The first linear layer has GELU activation function[35]. The embedding is projected to a higher dimension in the first layer and then mapped to its original size in the second layer. Supplementary Fig. 5b shows the structure of a MLP block in a Transformer Encoder.

The remaining internal embeddings are passed to a dropout layer followed by a 1D convolution layer for the gene prediction head. The gene prediction head uses a dropout layer and 1D convolution layer as the output layer similar to the HE2RNA

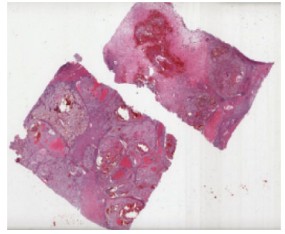

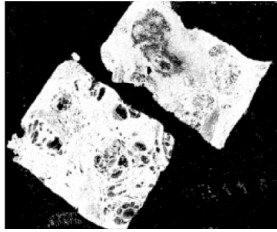

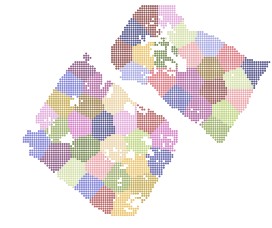

(a) Thumbnail of a WSI. (b) The tissue mask. (c) The *k*-means clusters.

**Fig. 4 An example of clustering for creating bag of tiles from a WSI. a** Shows a thumbnail of a WSI, **b** shows the tissue mask obtained by segmenting the WSI, and **c** shows the clustered WSI using k-means.

model introduced in ref. [5]. However, the first two layers, which were two 1D convolution layers responsible for feature extraction in HE2RNA, were replaced with a Transformer Encoder to capture the relationship between all instances. As the model produces one prediction per gene per instance, the same aggregation strategy described in ref. [5] was adapted for computing the gene prediction for each WSI. In particular, Schmauch et al. sampled a random number $n$ at each iteration and calculated each gene's prediction by averaging the top-$n$ predictions by tiles in a WSI (bag)[5]. They suggested this approach acts as a regularization technique and decreases the chance of overfitting[5]. As there were 49 tile embeddings in each bag, $n$ was randomly selected from {1,2,5,10,20,49}. For a randomly selected $n$ during training, the gene prediction outcome can be written as

$$\mathbf{s} = \text{Conv1D}(\mathbf{z}_L^{1:\text{end}}), \tag{9}$$

$$\mathbf{S}(n) = \sum_{i=1}^{n} \frac{\mathbf{s}^i}{n}, \tag{10}$$

where $\mathbf{z}_L^{1:\text{end}} \in \mathbb{R}^{D \times k}$, $\mathbf{s} \in \mathbb{R}^{D \times k}$, and $\mathbf{S}(n) \in \mathbb{R}^{d_g}$ are the internal embeddings excluding the class token, the tile-wise gene prediction, and slide-level gene expression prediction, respectively. During the test the final prediction $\mathbf{S}$ is calculated as an average of all possible values for $n$ as

$$\mathbf{S} = \sum_{i=1}^{k} \frac{\mathbf{S}(i)}{i}. \tag{11}$$

The mean squared error loss function is employed to learn gene predictions. Finally, the total loss for tRNAsformer is computed as

$$\mathcal{L}_{\text{Total}}(\theta) = \mathcal{L}_{\text{classification}}(\theta) + \gamma \mathcal{L}_{\text{prediction}}(\theta) + \lambda \mathcal{L}_{\text{regularization}}(\theta), \tag{12}$$

$$= \frac{1}{B}\sum_{i=1}^{B}(-\mathbf{y}_i \log(\hat{y}_i) + \gamma|\mathbf{y}_i^g - S_i|) + \lambda|\theta|_2^2, \tag{13}$$

where $\theta, \lambda, \gamma, B, \mathbf{y}^g$ are the model parameters, weight regularization coefficient, hyperparameter for scaling the losses, number of samples in a batch, and true bulk RNA-seq associated with the slides. A summary of the proposed approach is included in Fig. 1.

**Training settings for training tRNAsformer models**. To begin, TCGA cases are split into 80%, 10%, and 10% subsets for the training, validation, and test sets. Each case was associated with a patient and could have contained multiple diagnostic WSIs or RNA-seq files. During the training process, the number of bags has been considered as a hyper-parameter to optimize the model performance. After optimizing the hyper-parameters, 100 bags were sampled from each WSI. As a result, the training set comprised of 63,400 bags (see Supplementary Table 1).

The tRNAsformer's internal representation size was set to 384. The MLP ratio and the number of self-attention heads were both four. The tRNAsformer was trained for 20 epochs with a minibatch of size 64. The AdamW was chosen as the optimizer with a starting learning rate of $3 \times 10^{-4}$ [36]. The weight regularization coefficient was set to 0.01 to avoid overfitting. The reduce-on-plateau method was chosen for scheduling the learning rate. Therefore, the learning rate was reduced by ten every two epochs without an improvement in the validation loss. The scaling coefficient $\gamma$ was set to 0.5. The last dropout layer's probability was set to 0.25. The values for the model with the lowest validation loss are reported. All experiments are conducted using a single NVIDIA GeForce RTX 2080 SUPER graphic card. The desktop's CPU was Intel(R) Core(TM) i9-10900X.

**Training settings for training MLP model**. Another model was trained based on the MLP architecture, called HE2RNA, described in ref. [5]. The trained HE2RNA model was not provided by the authors of the HE2RNA paper. Therefore, we have built and trained the HE2RNA model using the same dataset used for training tRNAsformer such that we produce a fair benchmark based on current literature. The fully connected layers were replaced with successive 1D convolutions with kernel size one and stride one to slide data due to practicality in the MLP design[5]. A dropout layer is applied between successive layers, and the activation function was ReLU. The model based on MLP design suggested in ref. [5] is referred to as HE2RNA$_{\text{bb}}$ (bb stands for backbone) as it was trained on TCGA training set used in this paper. The HE2RNA Rb$_b$ model is made of three 1D convolutional layers. The first two layers each contained $h$ input and output channels, whereas the last layer had the same number of output channels as the number of genes. In other words, $h$ is the size of the model's internal representation. The $h$ was set to 1024 for HE2RNA$_{\text{bb1024}}$. The model was trained for 20 epochs using AdamW optimizer and a starting learning rate of $3 \times 10^{-4}$ [36]. If no improvement is observed for the validation loss for two epochs, the learning rate was reduced by ten. The minibatch size was set to 64. The values for the model with the lowest validation loss are provided. The number of parameters of each model is shown in Supplementary Table 2 for comparison. The wall clock time for a single epoch for training and validation is also provided in the same table as the number of parameters.

**The Ohio State University kidney dataset**. This is an internal dataset that we used to evaluate the internal representation of our model. The pathology department's surgical pathology files were examined for consecutive cases of renal cell

carcinoma classified as clear cell carcinoma (ccRCC), chromophobe renal cell carcinoma (crRCC), or papillary renal cell carcinoma (pRCC). The dataset was created at the end of the search, and it contained 142 instances of renal cell carcinoma. The WSIs from ccRCC, crRCC, and pRCC were 48, 44, and 50, respectively. Each patient had one representative cancer slide that was examined by a board-certified pathologist (Anil V. Parwani) before being scanned at 20 × utilising an aperio XT scanscope (Leica biosystems, CA). A boardcertified pathologist (AP) reviewed the WSI images and validated the classifications a second time to guarantee the image quality and correctness of the diagnosis.

**External validation of the transcriptomic learning for representing WSIs**. The model that was trained on the TCGA kidney dataset was used to embed the external dataset. The classification and WSI search studies were then performed to examine domain change impact on the proposed pipeline.

**Reporting summary**. Further information on research design is available in the Nature Portfolio Reporting Summary linked to this article.

## Data availability
The NCI Genomic Data Commons Portal (https://portal.gdc.cancer.gov) has all of the TCGA digital slides available to the public. For reproducibility, the processed data of each case obtained from the TCGA project are available at https://doi.org/10.5281/zenodo.7613408. The data includes a csv file for each case, which lists all the 31,793 gene expression scores that we have considered in our experiments.

## Code availability
Our source code along with the trained tRNAsformer models are available at https://doi.org/10.5281/zenodo.7613349.

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

# ARTICLE

15. Tabibu, S., Vinod, P. & Jawahar, C. Pan-renal cell carcinoma classification and survival prediction from histopathology images using deep learning. *Sci. Rep.* **9**, 10509 (2019).

16. Dietterich, T. G., Lathrop, R. H. & Lozano-Pérez, T. Solving the multiple instance problem with axis-parallel rectangles. *Artif. Intell.* **89**, 31–71 (1997).

17. Hubbard, T. et al. The ensembl genome database project. *Nucleic Acids Res.* **30**, 38–41 (2002).

18. Bartha, Á. & Győrffy, B. Tnmplot. com: a web tool for the comparison of gene expression in normal, tumor and metastatic tissues. *Int. J. Mol. Sci.* **22**, 2622 (2021).

19. Luo, M.-S., Huang, G.-J. & Liu, B.-X. Immune infiltration in nasopharyngeal carcinoma based on gene expression. *Medicine* **98**, e17311 (2019).

20. Hoffman, G. E. & Schadt, E. E. variancepartition: interpreting drivers of variation in complex gene expression studies. *BMC Bioinforma.* **17**, 1–13 (2016).

21. Campbell-Staton, S. C., Velotta, J. P. & Winchell, K. M. Selection on adaptive and maladaptive gene expression plasticity during thermal adaptation to urban heat islands. *Nat. Commun.* **12**, 1–14 (2021).

22. Avsec, Ž. et al. Effective gene expression prediction from sequence by integrating long-range interactions. *Nat. Methods* **18**, 1196–1203 (2021).

23. Holm, S. A simple sequentially rejective multiple test procedure. *Scand. J. Stat.* **6**, 65–70 (1979).

24. Benjamini, Y. & Hochberg, Y. Controlling the false discovery rate: a practical and powerful approach to multiple testing. *J. R. Stat. Soc.: Ser. B* **57**, 289–300 (1995).

25. Spyromitros-Xioufis, E., Tsoumakas, G., Groves, W. & Vlahavas, I. Multi-target regression via input space expansion: treating targets as inputs. *Mach. Learn.* **104**, 55–98 (2016).

26. Stacke, K., Eilertsen, G., Unger, J. & Lundström, C. A closer look at domain shift for deep learning in histopathology. Preprint at https://arxiv.org/abs/1909.11575 (2019).

27. Asilian Bidgoli, A., Rahnamayan, S., Dehkharghanian, T., Grami, A. & Tizhoosh, H. Bias reduction in representation of histopathology images using deep feature selection. *Sci. Rep.* **12**, 1–12 (2022).

28. Dehkharghanian, T. et al. Biased data, biased AI: deep networks predict the acquisition site of TCGA images. *BMC Diagnostic Pathology* (2023).

29. Safarpoor, A., Shafiei, S., Gonzalez, R., Parwani, A. & Tizhoosh, H. Renal cell carcinoma whole-slide image classification and search using deep learning. *Research Square* https://doi.org/10.21203/rs.3.rs-971708/v1 (2021).

30. Kalra, S. et al. Yottixel-an image search engine for large archives of histopathology whole slide images. *Med. Image Anal.* **65**, 101757 (2020).

31. Dehghani, M., Zamani, H., Severyn, A., Kamps, J. & Croft, W. B. Neural ranking models with weak supervision. in *Proceedings of the 40th International ACM SIGIR Conference on Research and Development in Information Retrieval* 65–74 (2017).

32. Sikaroudi, M. et al. Supervision and source domain impact on representation learning: a histopathology case study. in *2020 42nd Annual International Conference of the IEEE Engineering in Medicine & Biology Society (EMBC)* 1400–1403 (IEEE, 2020).

33. Gildenblat, J. & Klaiman, E. Self-supervised similarity learning for digital pathology. Preprint at https://arxiv.org/abs/1905.08139 (2019).

34. Dosovitskiy, A. et al. An image is worth $16 \times 16$ words: Transformers for image recognition at scale. Preprint at https://arxiv.org/abs/2010.11929 (2020).

35. Hendrycks, D. & Gimpel, K. Gaussian error linear units (gelus). Preprint at https://arxiv.org/abs/1606.08415 (2016).

36. Loshchilov, I. & Hutter, F. Decoupled weight decay regularization. Preprint at https://arxiv.org/abs/1711.05101 (2017).

## Acknowledgements
This project was partially funded as part of an ORF-RE consortium by the Government of Ontario.

## Author contributions
A.A. has contributed to conception of main ideas, restructured the paper, re-analyzed the data, and revised the manuscript. A.S. contributed to and discussed initial ideas, designed and performed the initial experiments, analyzed and interpreted the results, and wrote the first draft. M.S. contributed to data processing and analysis. H.R.T. conceived the initial idea, oversaw the entire study, analyzed the data/results, and wrote parts of the paper. J.D.H. contributed to the project management, revised the paper and provided critical feedback.

## Competing interests
The authors declare no competing interests.

## Ethics approval and consent to participate
This study was approved by the Ohio State University institutional research board. Informed written consent was obtained from all individual patients included in the study. All methods were carried out in accordance with relevant guidelines and regulations. All the data was de-identified using an honest broker system.
