## [Peer Review File · Communications Biology]

Reviewers' comments:

Reviewer #1 (Remarks to the Author):

Revision for COMMSBIO-22-0953-T

In this study, the authors combine cancer imaging and molecular fingerprinting to train an AI model for fast classification of cancer subtypes. The authors make a compelling argument that since survival rates differ across RCC subtypes, the fast identification of subtype based on WSI and prediction of gene expression could benefit prognosis and diagnosis. As described by authors, the gathering and stratification of data for training, testing, validation (from TCGA), and the use of an external dataset for validation of WSI representation (from Ohio State University), make this a well controlled and organized effort in AI model training and generalization.

While the study is clearly relevant for the field and presents an innovative take in the union of biological data types (morphology and molecular fingerprint) for training of predictive AI models, the authors fail to provide sufficient data and detail for the study to be reproducible. Therefore, I believe the manuscript requires revisions before being accepted for publication.

Minor issues:

Authors could help this manuscript reach a broader readership by defining and clarifying acronyms and jargon such as RCC, attention-based topology, weakly supervised, "pixel-level annotation", P@K, AP@K, MAP@K, among others.

Figures 6, 7, 8 and 9 are referenced before other main text figures.

Major issues:

Since this article bears striking resemblance to the 2020 Nature Communications article "A deep learning model to predict rna-seq expression of tumours from whole slide images.", which the authors here use to benchmark their own model, a special effort should be made to highlight the differences between the two studies. Beyond the introduction of a transformer layer, where there other significant differences between HE2RNA and tRNAsformer?

It is not clear if the authors created a new model based on the description of HE2RNA provided in the above mentioned paper, or if the authors used the pre-trained HE2RNA model described by that group. Please clarify if the "HE2RNA" model was trained by the authors or obtained from the 2020 Nature Communications publication.

Authors indicate that "Another important observation is that tRNAsformer has higher mean correlation coefficients than its counterpart for $L = 2$ to $L = 8$." The difference in mean correlation coefficient R between tRNAsformer ($L=9$) and HE2RNA is:

$$0.335 - 0.315 = 0.02$$

Which is well within the standard deviation of the correlation coefficients (0.185 and 0.176, respectively). The same is true for Spearman's correlation coefficients.

How do authors justify the statement that "tRNAsformer has higher mean correlation coefficients than its counterpart" since they seem to be statistically indistinguishable?

Given the results in Table 2 for MAE, RMSE, and RRMSE, where HE2RNA produces smaller errors than tRNAsformer, and the observation in Table 1 where tRNAsformer captures a larger number of genes with statistically significant changes in expression, could it be argued that tRNAsformer produces a better "qualitative" overview of gene expression, while HE2RNA produces a more precise single gene prediction value?

The authors state (when talking about their model) that "It can also predict gene expressions from H&E slides better than other methods [5]." I think the claim may be overstated due to the error estimation results mentioned above.

While the authors do indicate the address to the web portal (<https://portal.gdc.cancer.gov/>) from

which data was retrieved, they should also indicate the exact dataset used in this study, to assure reproducibility of the results. Listing the properties of selected cases is insufficient since the database may change over time, adding or removing entries used in this study.

If retaining and sharing the images and transcription data is impractical, authors should provide a comprehensive list of all cases (Case IDs) and projects these cases belonged to, and the exact data gathered from these cases, so that other may reconstruct the dataset as utilized in this study.

Authors should also improve the detailing of any pre-processing used when preparing the data for model training. For example, in the "Gene expression preprocessing" section, authors indicate that thousands of genes were excluded from the analysis. Please add the list of 31,793 genes actually used in this study in an SI file.

Would the log10-transformed values for gene expression be too large for release in an SI file as well?

Reviewer #2 (Remarks to the Author):

The paper introduced tRNAsformer, a transformer-based gene prediction method using whole slide images.

The problem trying to solve is interesting and the method and results are well presented.

Most part of the paper can be acceptable, especially Transcriptomic learning for WSI representation part is good, however, there are some concerns for publication.

1. It is not clearly stated that which model is the best model among L=1 to 12, though L=2 to 8 are comparable. It is not friendly for readers and users.
2. In section "A model for predicting gene expression from WSIs.," the correlation between what and what is not clear at first. I found it in the description in Figure 1, but it should be mentioned in the main text.
3. In section "A model for predicting gene expression from WSIs.," the differences to the existing method (HE2RNA_bb1024) is very slight or sometimes worse than existing method. I don't think this paper should be rejected just because of this because other part are well outperform to existing method, but the authors should conclude honestly.
4. Will the pretrained models be available on your website? Only the source code is mentioned in the availability section.

Reviewer #3 (Remarks to the Author):

By integrating bulk RNA-sequencing data and H&E-stained whole-slide images from the TCGA database, Safarpour and co-authors presented a deep-learning model, tRNAsfomer, to predict the gene expression from H&E slides. And tRNAsfomer was used to address the classification and searching problem of weakly annotated H&E slides. This is an interesting topic, but some concerns need to be addressed before publication.

1. The details of obtaining the raw data from the TCGA database need to be disclosed. And the description of these data needs to be more clear. As is shown in the manuscript, the source codes are currently not available for a proper evaluation by the reviewers.
2. Since the metadata of the samples from the TCGA database is not available from the current manuscript, how many original projects/experiments were involved in these data is not clear. And the output of the model for RNAseq data prediction is the log-transformed FPKM-UQ data, which currently does not consider the batch effects among different projects. The mean value of FPKM-UQ data of each gene may vary dramatically between different projects. Thus, this would make the

correlation coefficient as an evaluation metric less reliable. This conclusion can be seen in Table 2 – the RRMSE value of HE2RNAbb1024 was even lower (better) than that of the best tRNAsformer model, i.e., tRNAsformerL=4. In addition, considering the expression of genes well predicted by a statistically significant correlation with a p-value < 0.01 after HS and BH correction is a very loose criterium. When p-value = 0.01, what is the value of R? Moreover, given this loose criterium, the difference in the numbers of well-predicted genes by tRNAsformerL=4 and by HE2RNAbb1024 is so trivial (with a 0.6% of improvement). Of note, at this moment the hyperparameters of HE2RNAbb1024 were even not optimized. So, it is hard to convince the readers that tRNAsformer is better than HE2RNA.

3. The mean correlation coefficient of the best model is only 0.341 ± 0.16 , how good is this result? Should you provide evidence by showing several scatter plots of the predicted RNAseq data vs real RNAseq data?

4. According to Table 3, the performance of the models decreases a lot when it applies to the external dataset. This makes me wonder how sensitive is the model to the color/exposure of the H&E-stained images, as different experimenters may get darker/lighter images even from the same specimen? Can this sensitivity be reduced by more data augmentations, such as color modification, flipping, cropping, rotation, noise injection, and random erasing? As is shown in the manuscript, 100 bags were created from each TCGA WSI, have you considered the number of bags as a hyper-parameter to optimize it?

5. In many experiments, the H&E-stained slide is only a very thin slice of a specimen, but RNAseq profiles a much bigger specimen which is not the same as the H&E-stained one. And one of the important features of the tumor specimen is its heterogeneous compositions/cells, which has been shown by many spatial transcriptomic studies. In other words, the RNAseq data may not directly correspond to the H&E-stained slide. Thus, can you verify the model using the spatial transcriptomic data in which both the RNAseq profiling and H&E staining are performed on the same slice of the specimen?

6. How the search experiments were performed was not properly described. Please detail this section.

Minor points:

1. Please add line numbers to the manuscript to make it easier to refer to.
2. There are some grammar mistakes, please correct them.
3. In the section "Transcriptomic learning for WSI representation – WSI classification," you showed two-dimensional PCA projections in Fig.7&9 without proper explanation, what is the purpose of showing these two figures?

All **changes** in the revised manuscript are **highlighted in BLUE** color.

Response to Reviewers' comments

Reviewer #1 (Remarks to the Author):

Revision for COMMSBIO-22-0953-T

In this study, the authors combine cancer imaging and molecular fingerprinting to train an AI model for fast classification of cancer subtypes. The authors make a compelling argument that since survival rates differ across RCC subtypes, the fast identification of subtype based on WSI and prediction of gene expression could benefit prognosis and diagnosis. As described by authors, the gathering and stratification of data for training, testing, validation (from TCGA), and the use of an external dataset for validation of WSI representation (from Ohio State University), make this a well controlled and organized effort in AI model training and generalization.

While the study is clearly relevant for the field and presents an innovative take in the union of biological data types (morphology and molecular fingerprint) for training of predictive AI models, the authors fail to provide sufficient data and detail for the study to be reproducible. Therefore, I believe the manuscript requires revisions before being accepted for publication.

Minor issues:

Authors could help this manuscript reach a broader readership by defining and clarifying acronyms and jargon such as RCC, attention-based topology, weakly supervised, “pixel-level annotation”, P@K, AP@K, MAP@K, among others.

Thank you for this comment. The acronyms and jargon have been clarified in the revised manuscript, as described below:

The following sentence has been **modified** in the **Introduction section** to clarify RCC terminology: “The renal cell carcinoma (RCC) is the most common kidney cancer that is responsible for 85% of malignant cases [14].”

We also **added** the following paragraph to the **Introduction section** to clarify the attention-based topology:

“Our model employs transformer modules built on the attention mechanism to gather information required for learning WSI representations. The attention-based mechanism allows learning information that attributes to some specific features in the image and

scores them against other features. In doing so, the model would capture how a feature relates to the others in the image so that it focuses on the relevant part of the image.”

The following paragraph has been **added to the Discussion section** to make weakly supervised learning and pixel-level annotation clear for the reader:

“In fact, whole slide images (WSIs) are usually labeled by treating the image in its entirety (label is assigned to the whole image). For example, a whole slide image can be labeled as a tumor slide although it might include some normal tissue as well. Processing an entire WSI at once is not possible with present hardware technology. These images are commonly divided into smaller, more manageable pieces known as patches or tiles. However, large WSI datasets are generally softly labeled since pixel-level expert annotation is costly and labour-intensive. As a result, some of the tiles may not carry information that is relevant to the diagnostic label associated with the WSI. tRNAsformer design allowed for more efficient and precise processing of a collection of samples. It employs a weakly supervised learning along with multi-instance learning (MIL) concept. Weakly supervised learning is an approach to train a deep network by using the combination of the given labeled data and the weak supervision for obtaining new labeled data. This approach makes training a deep network possible when the available labeled data is insufficient. Furthermore, tRNAsformer employs the concept of MIL to handle the problem of having the real gene expression values per WSI instead of per tile. MIL is a form of weakly supervised learning where training instances are arranged in bags (sets), and a label is provided for the entire bag.”

In the **Results section**, we clarified P@K, AP@K, and MAP@K by adding the following:

“Later, P@K (Precision@K) and AP@K (Average Precision@K) were computed for each subset. P@K reflects how many relevant images are present in the top-k recommendations that the model suggests, while AP@K is the mean of P@i for $i=1, \dots, K$. Finally, the MAP@K (Mean Average Precision@K) value was computed by taking the average of 100 queries associated with 100 search subsets.”

Figures 6, 7, 8 and 9 are referenced before other main text figures.

Thank you for pointing this out. We have **modified** this in the revised manuscript so that the sequence of the figures is consistent with the referenced sequence in the text.

Major issues:

Since this article bears striking resemblance to the 2020 Nature Communications article “A deep learning model to predict rna-seq expression of tumours from whole slide images.”, which the authors here use to benchmark their own model, a special effort should be made to highlight the differences between the two studies. Beyond the introduction of a transformer layer, where there other significant differences between HE2RNA and tRNAsformer?

Thank you for the insightful comments. We agree that highlighting the differences between the two models from different aspects increases the manuscript quality. Therefore, the following **two paragraphs have been added** to the **Discussion section** to more pronouncedly highlight differences between tRNAsformer and HE2RNA models.

“However, one has to bear in mind that tRNAsformer is a multi-task computational pathology tool that can be used not only for gene expression prediction, but also for learning WSI representation based on the tissue morphology and the molecular fingerprint of a biopsy sample, which can be integrated into image search and classification.”

“However, producing a gene expression score per tile, like HE2RNA, results in ignoring the dependencies between the tiles of a WSI as the real values are per WSI not per tile. tRNAsformer solves this issue by treating a WSI in its entirety and therefore producing one prediction per WSI. The model employs the concept of multi-instance learning to handle the problem of having the real gene expression values per WSI instead of per tile. Furthermore, from a computational point of view, considering all tiles to train the network is prohibitively time and resource-consuming as a single WSI can easily have several thousands tiles. Therefore, in tRNAsformer we addressed this issue by incorporating the attention mechanism and multiple instance learning concept in the training process.”

It is not clear if the authors created a new model based on the description of HE2RNA provided in the above mentioned paper, or if the authors used the pre-trained HE2RNA model described by that group. Please clarify if the “HE2RNA” model was trained by the authors or obtained from the 2020 Nature Communications publication.

Thank you for this comment. This point has been **clarified** in the **Methods section** by adding the following:

“Another model was trained based on the MLP architecture, called HE2RNA, described in [5]. The trained HE2RNA model was not provided by the authors of the HE2RNA paper. Therefore, we have built and trained the HE2RNA model using the same dataset used for training tRNAsformer such that we produce a fair benchmark based on current literature.”

Authors indicate that “Another important observation is that tRNAsformer has higher mean correlation coefficients than its counterpart for L = 2 to L = 8.” The difference in mean correlation coefficient R between tRNAsformer (L=9) and HE2RNA is:

$$0.335 - 0.315 = 0.02$$

Which is well within the standard deviation of the correlation coefficients (0.185 and 0.176, respectively). The same is true for Spearman’s correlation coefficients.

How do authors justify the statement that “tRNAsformer has higher mean correlation coefficients than its counterpart” since they seem to be statistically indistinguishable?

Thank you for this comment. We agree that the description of the correlation results was misleading. Therefore, we **modified** this in the revised **Results section** to make it more understandable from the reader point of view, which now reads “In terms of the correlation of the predicted gene expressions with real values, tRNAsformer models from L=2 to L=8 achieved comparable results with a slight improvement as compared to HE2RNA.”

We also **added** the following paragraph to the **Discussion section**:

“tRNAsformer was able to predict gene expression scores with a slightly improved correlation compared to that achieved by HE2RNA. However, one has to bear in mind that tRNAsformer is a multi-task computational pathology tool that can be used not only for gene expression prediction, but also for learning WSI representation based on the tissue morphology and the molecular fingerprint of a biopsy sample, which can be integrated into image search and classification. The correlation metric was used to evaluate only one task, which is the gene expression prediction. The other task (i.e., transcriptomic learning for WSI representation for image search and classification) has been evaluated by considering an external dataset along with two other methods for comparison, namely “Yottixel” and “Low power” methods.”

We also **highlighted** the purpose from each experiment by **adding** the following paragraph at the beginning of the **Results section**.

“In this section, we assess the performance of tRNAsformer in terms of the two main tasks that it has been trained for: gene expression prediction from WSI and WSI representation for image search and classification. The performance of tRNAsformer in predicting gene expressions has been compared with the performance of one of a state-of-the-art model, called HE2RNA. The performance of tRNAsformer in terms of learning rich information to represent WSIs has been compared to two other methods, namely Yottixel and Low Power.”

Given the results in Table 2 for MAE, RMSE, and RRMSE, where HE2RNA produces smaller errors than tRNAsformer, and the observation in Table 1 where tRNAsformer captures a larger number of genes with statistically significant changes in expression, could it be argued that tRNAsformer produces a better “qualitative” overview of gene expression, while HE2RNA produces a more precise single gene prediction value?

We thank you for your insightful comment. This point has been **clarified** in the **Results section** by adding the following:

“HE2RNA uses all the tiles of a WSI to train the model and to produce a prediction for every tile. This helps in improving the error rate when averaging a large number of tile predictions to get one prediction per slide. Averaging multiple predicted values (tile predictions) would increase the chance of having a more similar value to the real value as the effect of applying this method is like averaging the error rate of all the predictions to get a single representative value of all the tiles. However, producing a gene expression score per tile, like HE2RNA, results in ignoring the dependencies between the tiles of a WSI as the real values are per WSI not per tile. tRNAsformer solves this issue by treating a WSI in its entirety and therefore producing one prediction per WSI. The model employs the concept of multi-instance learning to handle the problem of having the real gene expression values per WSI instead of per tile. Furthermore, from a computational point of view, considering all the tiles to train the network is prohibitively time and resource-consuming as a single WSI can easily have several thousands tiles. Therefore, in tRNAsformer we addressed this issue by incorporating the attention mechanism and multiple instance learning concept in the training process.”

The authors state (when talking about their model) that “It can also predict gene expressions from H&E slides better than other methods [5].” I think the claim may be overstated due to the error estimation results mentioned above.

Thank you for pointing this out. We agree that this statement can be misleading. Therefore, we have **modified** the sentence in the **Discussion section** of the revised manuscript, which now reads: “It can also predict gene expressions from H&E slides with a comparable performance with some improvement as compared to other state-of-the-art methods [5].”

While the authors do indicate the address to the web portal (<https://portal.gdc.cancer.gov/>) from which data was retrieved, they should also indicate the exact dataset used in this

study, to assure reproducibility of the results. Listing the properties of selected cases is insufficient since the database may change over time, adding or removing entries used in this study.

If retaining and sharing the images and transcription data is impractical, authors should provide a comprehensive list of all cases (Case IDs) and projects these cases belonged to, and the exact data gathered from these cases, so that other may reconstruct the dataset as utilized in this study.

Thank you for drawing our attention to this important point. We agree that providing a **detailed list of all cases** is important for reproducibility. Hence, we will **provide a csv file** with information about all the cases we obtained from the TCGA project. We also **clarified** this in the **Data Availability section** by adding the following paragraph:

“The NCI Genomic Data Commons Portal (<https://portal.gdc.cancer.gov>) has all of TCGA digital slides available to the public. For reproducibility, a csv file with information about all the cases that we acquired from the TCGA project will be made publicly available on our lab’s website (<https://kimia.uwaterloo.ca>).”

Authors should also improve the detailing of any pre-processing used when preparing the data for model training. For example, in the “Gene expression preprocessing” section, authors indicate that thousands of genes were excluded from the analysis. Please add the list of 31,793 genes actually used in this study in an SI file.

Would the log10-transformed values for gene expression be too large for release in an SI file as well?

Thank you for this comment. We think this is important for reproducibility. Following the reviewer's suggestion, we will **provide a csv file** that details all the information about the cases we obtained from the TCGA database, which makes it possible for other researchers to obtain both WSIs and RNA-seq files of all the cases from TCGA. This has been mentioned in the **Data Availability section** (as described above).

We will also **provide a csv file for each case**, which lists all the **31,793 gene expression scores** that we have considered in our experiments. We **mentioned** this in the **Data Availability section** by adding the following:

“We will also provide a csv file for each case, which lists all the 31,793 gene expression scores that we have considered in our experiments.”

Reviewer #2 (Remarks to the Author):

The paper introduced tRNAsformer, a transformer-based gene prediction method using whole slide images.

The problem trying to solve is interesting and the method and results are well presented.

Most part of the paper can be acceptable, especially Transcriptomic learning for WSI representation part is good, however, there are some concerns for publication.

1. It is not clearly stated that which model is the best model among L=1 to 12, though L=2 to 8 are comparable. It is not friendly for readers and users.

Thank you for this comment. As you mentioned, the performance of models with L=2 to 8 are comparable. Therefore, we **clarified** this by adding the following paragraph to the **Results section**.

“Overall, as it can be noted from the above results, the performance of tRNAsformer models with L=2 to 8 are comparable. However, by considering all the metrics used to evaluate the models, tRNAsformer with L=4 performs the best. In this paper, we are presenting the performance of tRNAsformer with different depth as the model depth can be selected based on the available resources. For example, in the case of limited resources, L=2 can be used as it can achieve a comparable performance as the deeper models but with less resource requirement.”

2. In section "A model for predicting gene expression from WSIs.," the correlation between what and what is not clear at first. I found it in the description in Figure 1, but it should be mentioned in the main text.

Thank you for pointing this out. We have **clarified** this in the **Results section** of the revised manuscript, as follows:

“Fig. 1 illustrates the distribution of the correlation coefficients for 31,793 genes predicted by different models along with their true values in the test set of TCGA.”

3. In section "A model for predicting gene expression from WSIs.," the differences to the existing method (HE2RNA_bb1024) is very slight or sometimes worse than existing method.

I don't think this paper should be rejected just because of this because other part are well outperform to existing method, but the authors should conclude honestly.

Thank you for this comment. We agree that the conclusion of the results may be misleading. We have **modified** this in the revised **Results section** so that the difference between the performance of the two models is described clearly, which now reads: ““In terms of the correlation of the predicted gene expressions with real values, tRNAsformer models from L=2 to L=8 achieved comparable results with a slight improvement as compared to HE2RNA.”

We also **added** the following two paragraphs to the **Discussion section** so that the conclusion of the comparison between tRNAsformer and HE2RNA models is more reasonable for the reader.

“tRNAsformer was able to predict gene expression scores with a slightly improved correlation compared to that achieved by HE2RNA. However, one has to bear in mind that tRNAsformer is a multi-task computational pathology tool that can be used not only for gene expression prediction, but also for learning WSI representation based on the tissue morphology and the molecular fingerprint of a biopsy sample, which can be integrated into image search and classification. The correlation metric was used to evaluate only one task, which is the gene expression prediction.”

We also **added** the following to the **Discussion section**:

“HE2RNA uses all the tiles of a WSI to train the model and to produce a prediction for every tile. This helps in improving the error rate when averaging a large number of tile predictions to get one prediction per slide. Averaging multiple predicted values (tile predictions) would increase the chance of having a more similar value to the real value as the effect of applying this method is like averaging the error rate of all the predictions to get a single representative value of all the tiles. However, producing a gene expression score per tile, like HE2RNA, results in ignoring the dependencies between the tiles of a WSI as the real values are per WSI not per tile. tRNAsformer solves this issue by treating a WSI in its entirety and therefore producing one prediction per WSI. The model employs the concept of multi-instance learning to handle the problem of having the real gene expression values per WSI instead of per tile. Furthermore, from a computational point of view, considering all the tiles to train the network is prohibitively time and resource-consuming as a single WSI can easily have several thousands tiles. Therefore, in tRNAsformer we addressed this issue by incorporating the attention mechanism and multiple instance learning concept in the training process.”

4. Will the pretrained models be available on your website? Only the source code is mentioned in the availability section.

Thank you for this comment. The tRNAsfomer model **will be available along with the source code on our website**. We mentioned this in the **Code Availability section** of the revised manuscript by adding the following: “Upon publishing, our source code along with the trained tRNAsfomer models will be made publicly available on our lab’s website (<https://kimia.uwaterloo.ca>).”

Reviewer #3 (Remarks to the Author):

By integrating bulk RNA-sequencing data and H&E-stained whole-slide images from the TCGA database, Safarpour and co-authors presented a deep-learning model, tRNAsfomer, to predict the gene expression from H&E slides. And tRNAsfomer was used to address the classification and searching problem of weakly annotated H&E slides. This is an interesting topic, but some concerns need to be addressed before publication.

1. The details of obtaining the raw data from the TCGA database need to be disclosed. And the description of these data needs to be more clear. As is shown in the manuscript, the source codes are currently not available for a proper evaluation by the reviewers.

Thank you for this comment. We think this is important for reproducibility. Following the reviewer's suggestion, we will **provide a csv file** that details all the information about the cases we obtained from the TCGA database. We mentioned this in the Data Availability section of the revised manuscript, which now reads: “the NCI Genomic Data Commons Portal (<https://portal.gdc.cancer.gov>) has all of the TCGA digital slides available to the public. For reproducibility, a csv file with information about all the cases that we acquired from TCGA project will be made publicly available on our lab’s website (<https://kimia.uwaterloo.ca>). We will also provide a csv file for each case, which lists all the 31,793 gene expression scores that we have considered in our experiments.”

We have also **clarified** the description of the obtained data in the **Methods section** of the revised manuscript by adding the following:

“The data used in this study was obtained from TCGA (<https://portal.gdc.cancer.gov/>). Three kidney datasets have been considered, which are TCGA-KIRC, TCGA-KIRP, and TCGA-KICH. Only cases that have WSI as well as RNAseq profile were considered. We

selected H&E-stained formalin-fixed, paraffin-embedded (FFPE) diagnostic slides. The retrieved cases included three subtypes, clear cell carcinoma, ICD-O 8310/3, (ccRCC), chromophobe type - renal cell carcinoma, ICD-O 8317/3, (crRCC), and papillary carcinoma, ICD-O 8260/3, (pRCC). For transcriptomic data, we utilized Fragments Per Kilobase of transcript per Million mapped reads upper quartile (FPKM-UQ) files. The detailed information regarding the cases are included in Table 5.”

2. Since the metadata of the samples from the TCGA database is not available from the current manuscript, how many original projects/experiments were involved in these data is not clear. And the output of the model for RNAseq data prediction is the log-transformed FPKM-UQ data, which currently does not consider the batch effects among different projects. The mean value of FPKM-UQ data of each gene may vary dramatically between different projects. Thus, this would make the correlation coefficient as an evaluation metric less reliable. This conclusion can be seen in Table 2 – the RRMSE value of HE2RNAbb1024 was even lower (better) than that of the best tRNAsformer model, i.e., tRNAsformerL=4. In addition, considering the expression of genes well predicted by a statistically significant correlation with a p-value < 0.01 after HS and BH correction is a very loose criterium. When p-value = 0.01, what is the value of R? Moreover, given this loose criterium, the difference in the numbers of well-predicted genes by tRNAsformerL=4 and by HE2RNAbb1024 is so trivial (with a 0.6% of improvement). Of note, at this moment the hyperparameters of HE2RNAbb1024 were even not optimized. So, it is hard to convince the readers that tRNAsformer is better than HE2RNA.

We thank you for detailed comments and appreciate insightful comments on the paper. As mentioned in the answer of the previous comment, we will **provide a csv file** with information about all the cases we obtained from the TCGA project. This has been **mentioned** in the **Data Availability section**, which now reads:

“The NCI Genomic Data Commons Portal (<https://portal.gdc.cancer.gov>) has all of TCGA digital slides available to the public. For reproducibility, a csv file with information about all the cases that we acquired from TCGA project will be made publicly available on our lab’s website (<https://kimia.uwaterloo.ca>).”

We agree that the mean value of the FPKM-UQ data for each gene may vary significantly between different projects. Therefore, we **clarified** the source of the data we used by adding the following to the revised **Methods section**:

“As the mean value of the FPKM-UQ data for each gene may vary significantly between different projects, both tRNAsformer and HE2RNA models have been evaluated to predict

gene expression scores of FPKM-UQ data from only one project, which is TCGA. Three kidney datasets have been considered from TCGA, which are TCGA-KIRC, TCGA-KIRP, and TCGA-KICH. Furthermore, we have excluded genes with a median expression of zero to improve the interpretability of the results.”

We will **provide a csv file** for each case, which lists all the 31,793 gene expression scores that we have considered in our experiments. We **mentioned** this in the **Data Availability section** by adding the following:

“We will also provide a csv file for each case, which lists all the 31,793 gene expression scores that we have considered in our experiments.”

We agree with the reviewer that the conclusion of the results were not reasonable enough to convince the reader about the performance of tRNAsformer as compared to HE2RNA. Therefore, we **added** the following **two paragraphs** to the revised **Results section** and **Discussion section, respectively**, so that the conclusion of the comparison between tRNAsformer and HE2RNA models is more reasonable for the reader.

“The hyperparameters of both tRNAsformer and HE2RNA models were optimized before conducting the experiments. HE2RNA uses all the tiles of a WSI to train the model and to produce a prediction for every tile. This helps in improving the error rate when averaging a large number of tile predictions to get one prediction per slide. Averaging multiple predicted values (tile predictions) would increase the chance of having a more similar value to the real value as the effect of applying this method is like averaging the error rate of all the predictions to get a single representative value of all the tiles. However, producing a gene expression score per tile, like HE2RNA, results in ignoring the dependencies between the tiles of a WSI as the real values are per WSI not per tile. tRNAsformer solves this issue by treating a WSI in its entirety and therefore producing one prediction per WSI. The model employs the concept of multi-instance learning to handle the problem of having the real gene expression values per WSI instead of per tile. Furthermore, from a computational point of view, considering all the tiles to train the network is prohibitively time and resource-consuming as a single WSI can easily have several thousands tiles. Therefore, in tRNAsformer we addressed this issue by incorporating the attention mechanism and multiple instance learning concept in the training process.”

“Our main aim from the comparison between tRNAsformer and HE2RNA is to demonstrate that tRNAsformer can predict gene expressions from a WSI as accurate as the state-of-the-art gene expression algorithms but with simultaneously learning rich WSI representation from both morphology features as well as molecular fingerprint, which can be used for applications such as image search. tRNAsformer was able to predict gene

expression scores with a slightly improved correlation compared to that achieved by HE2RNA. However, one has to bear in mind that tRNAsformer is a multi-task computational pathology tool that can be used not only for gene expression prediction, but also for learning WSI representation based on the tissue morphology and the molecular fingerprint of a biopsy sample, which can be integrated into image search and classification. The correlation metric was used to evaluate only one task, which is the gene expression prediction. The other task (i.e., transcriptomic learning for WSI representation for image search and classification) has been evaluated by considering an external dataset along with two other methods for comparison, namely “Yottixel” and “Low power” methods.”

3. The mean correlation coefficient of the best model is only 0.341 ± 0.16 , how good is this result? Should you provide evidence by showing several scatter plots of the predicted RNAseq data vs real RNAseq data?

Thank you for this comment. Although we agree that the scatter plots are generally, given the large number of values that would have to be scattered in the plot, this would not provide any easily visible clues. To address your comment we added the following paragraph in the Results section:

“Beyond correlation values, the literature uses violin plots [18-21] because the large number of data points per patient drastically reduces the visibility of any interpretable clues if other methods such as scatter plots [22] are used.”

4. According to Table 3, the performance of the models decreases a lot when it applies to the external dataset. This makes me wonder how sensitive is the model to the color/exposure of the H&E-stained images, as different experimenters may get darker/lighter images even from the same specimen? Can this sensitivity be reduced by more data augmentations, such as color modification, flipping, cropping, rotation, noise injection, and random erasing? As is shown in the manuscript, 100 bags were created from each TCGA WSI, have you considered the number of bags as a hyper-parameter to optimize it?

Thank you for this comment. We addressed your point by **adding** the following to the revised **Discussion section**:

“As shown in Table 3, the accuracy of tRNAsformer decreased by about 13% for the external validation. These results still show a reasonable performance especially when considering the performance of its counterpart, which showed about a 20% decrease in the accuracy. Lack of generalization due to overfitting, bias and shortcuts is a general problem in deep

learning [27, 28]. However, applying more sophisticated preprocessing may improve the model performance and lead to a better sensitivity when using an external dataset. The model performance can also be improved by training it on a larger dataset. However, for sake of reproducibility, we are limited to the number of WSIs available on TCGA. Furthermore, we can only consider WSIs where RNA-seq profiles were available in TCGA.”

We also **added** the following to the **Methods section**:

“During the training process, the number of bags has been considered as a hyper-parameter to optimize the model performance.”

5. In many experiments, the H&E-stained slide is only a very thin slice of a specimen, but RNAseq profiles a much bigger specimen which is not the same as the H&E-stained one. And one of the important features of the tumor specimen is its heterogeneous compositions/cells, which has been shown by many spatial transcriptomic studies. In other words, the RNAseq data may not directly correspond to the H&E-stained slide. Thus, can you verify the model using the spatial transcriptomic data in which both the RNAseq profiling and H&E staining are performed on the same slice of the specimen?

Thank you for this insightful comment. We agree that bulk RNA-seq data may be obtained from a different cut than the one used in the H&E slide, but very likely from the same tissue sample. We further agree that verifying the model using a special transcriptomic dataset where the H&E images and gene expression quantification performed on the same tissue section would add value to the manuscript. However, the available spatial transcriptomic datasets are very limited. tRNAsformer was trained using cases with kidney cancer. Therefore, a spatial transcriptomic dataset with kidney cancer cases is needed for further validation of the model. To the best of our knowledge, there is no publicly available spatial transcriptomic dataset on kidney cancer. We **added** this point as part of the future work at the end of the **Discussion section**:

“We have shown that even with RNA-Seq profiles obtained from bulk cells, mostly, isolated from different tissue section, tRNAsformer performed well in terms of predicting gene expression scores correlated to the true scores in the bulk RNA-seq profiles, which may indicate that most of the expressed genes in the tissue section used for H&E staining are also expressed in the tissue section used for RNA-Seq quantification. However, in future research, tRNAsformer can be investigated rigorously by verifying its performance using spatial transcriptomic data in which both the RNAseq profiling and H&E staining are performed on the same slice of the specimen.”

6. How the search experiments were performed was not properly described. Please detail this section.

Thank you for pointing this out. We **clarified** the search experiments in the **Results section**, which now reads:

“After training tRNAsformer, it was utilized to extract features (embeddings). To quantify the performance of tRNAsformer in WSI search, first, 100 subsets of instances were created from 8,000 TCGA test instances. Next, a pairwise distance matrix is computed using the WSI embeddings (feature vectors) for each subset. The Pearson correlation is employed as the distance metric. Following the leave-one-patient-out procedure, the top-k samples were determined for each instance (WSI). Later, P@K (Precision@K) and AP@K (Average Precision@K) were computed for each subset. P@K reflects how many relevant images are present in the top-k recommendations that the model suggests, while AP@K is the mean of P@i for $i=1, \dots, K$. Finally, the MAP@K (Mean Average Precision@K) value was computed by taking the average of 100 queries associated with 100 search subsets.”

“The performance of tRNAsformer has been compared with the performance of Yottixel [30], the state-of-the-art in WSI search, in terms of the mean average precision at different k, MAP@5 and MAP@10.”

Minor points:

1. Please add line numbers to the manuscript to make it easier to refer to.
2. There are some grammar mistakes, please correct them.
3. In the section “ Transcriptomic learning for WSI representation – WSI classification, ” you showed two-dimensional PCA projections in Fig.7&9 without proper explanation, what is the purpose of showing these two figures?

Thank you for drawing our attention to these points. Line numbers have been **added** to the revised manuscript. We have also **proofread** the manuscript so that the grammatical errors are **corrected**.

The purpose of showing Figures 7 & 9 has been **clarified** in the Results section, as follows:

“Fig. 7 & 9 show how well the WSI representations extracted from the tRNAsformer model can be distinguished across different classes. In other words, the figures illustrate the discriminative power of the features learnt by each tRNAsformer model.”

REVIEWERS' COMMENTS:

Reviewer #1 (Remarks to the Author):

The authors have addressed all concerns reasonably well and I believe the manuscript is ready for publication.

Reviewer #2 (Remarks to the Author):

The article gets much better with this revision. It might be acceptable for publishing.

Reviewer #3 (Remarks to the Author):

The authors have addressed my comments